

# PROTEVS-MED field experiments: Very High-Resolution Hydrographic Surveys in the Western Mediterranean Sea

Pierre Garreau[1], Franck Dumas[2], Stéphanie Louazel[2], Stéphanie Correard[2], Solenn Fercoq[2], Marc Le Menn[2], Alain Serpette[2], Valérie Garnier[1], Alexandre Stegner[3], Briac Le Vu[3], Andrea Doglioli[4], Gerald Gregori[4]

[1] IFREMER, Univ. Brest, CNRS UMR 6523, IRD, Laboratoire d'Océanographie Physique et Spatiale (LOPS), IUEM, 29280, Plouzané, France
[2] SHOM, Service Hydrographique et Océanographique de la Marine, 13 rue de Chatellier, CS592803, 29228 Brest CEDEX 2, France
[3] Laboratoire de Météorologie Dynamique (LMD), CNRS UMR 8539, Ecole Polytechnique, 91128 Palaiseau, France
[4] Aix Marseille Univ., Université de Toulon, CNRS, IRD, MIO UM 110, 13288 Marseille, France

*Correspondence to*: Pierre Garreau (pierre.garreau@ifremer.fr)

**Abstract:** From 2015 to 2018 four field experiments (7 legs) have been performed in the Western Mediterranean Basin during winter or early spring. The main objectives were the assessment of high-resolution modelling, the observation of mesoscale structure and associated ageostrophic dynamics. Thanks to the intensive use of a towed vehicle undulating in the upper oceanic layer between 0 and 400 meter depth (a SeaSoar), a large amount of very high resolution hydrographic transects (about 10.000 km) have been performed, observing mesoscale dynamics (slope current and its instabilities, anticyclonic eddies, sub-mesoscale coherent vortices, frontal dynamics convection events, strait outflows) and sub-mesoscale processes like stirring, mixed layer or symmetric instabilities. When available, the data were completed with velocities recorded by Vessel Mounted Acoustic Doppler Current Profiler (VMADCP) and by surface salinity and temperature recorded by ThermosalinoGraph (TSG). CTD casts have also been performed giving the background hydrography of the deeper layers when focusing on peculiar objects. In 2017, a free fall profiler (an MVP-200) has been deployed to manage even higher horizontal resolution. The aim of the survey was the dynamics and attention were paid to temperature, salinity and currents. Nevertheless, biological sensors (Chlorophyll *a*, Turbidity, Dissolved Oxygen etc.) have been opportunely carried out as they are able to provide complementary observations about the circulation. This data set is an unprecedented opportunity to investigate the very fine scale processes as the Mediterranean Sea is known for its intense and contrasted dynamics. It should be useful for modellers (who reduce the grid size below a few hundred meters) and expect to properly catch finer scale dynamics. Likewise, theoretical work could also be illustrated by in situ evidence





embedded in this data set. The data are available through SEANOE repository (https://doi.org/10.17882/62352; Dumas et al., 2018).

# 1 Introduction

Progress in numerical modelling and conceptual approaches both emphasized the importance of fine scale processes in connecting the ocean interior with the atmosphere, driving the energy cascade (McWilliams, 2016) and shaping the biochemical cycles and biodiversity distribution (Lévy et al., 2012; Lévy et al., 2018). For instance, Chorophyll a filaments near the external boundary of the North Current in the Ligurian Sea are generated by frontal instabilities (Niewiadomska et al., 2008) and coherent vortex may act efficiently both as biological barriers and drivers of plankton diversity (Bosse et al., 2017; Rousselet et al., 2019). As the scales interactions are ubiquitous, it is of crucial importance to develop observations strategy to get insight simultaneously of the large scale dynamics, the mesoscale and the sub-mesoscale processes. Unfortunately, it is not straightforward to reach this objective using conventional cruise strategies leading to a lack of in situ observations of fine scale processes. Due to their synoptic view, satellite observations fill partly the gap encompassing together large scale dynamics and finer structures. Remotely sensed observations of surface temperature, ocean colours or altimetry exhibit since many years a large spectrum of processes with various cut off scale (from around 70 km for the altimetry down to some tens of meters for imagery). Some of these limits will be pushed back soon: as for the future SWOT (Surface Water and Ocean Topography) satellite mission will; it is expected to provide substantial improvement for small scales processes having a sea surface height signature (d'Ovidio et al., 2019).

The data presented hereafter are a contribution to very high-resolution observations of the top oceanic layer and are freely available on SEANOE repository: https://doi.org/10.17882/62352 (Dumas et al., 2018). Long transects of the first 400 m depths under the surface was sampled with a horizontal resolution in the range of one nautical mile in the Western Mediterranean Sea. The hydrographic and dynamic background of this region is given in section 2. The objectives and implementation of the surveys are presented in section 3. The details of the measurements (vectors, sensors, methodology, metrology, data control, ancillary data) are reported in section 4. To illustrate the potentiality of the dataset, some quick looks of the observed processes are displayed in section 5. Lessons learned during surveys and summary are displayed in section 6.



## 2 Oceanic contexts in Western Mediterranean Sea

The Mediterranean Sea is often referred as a "pocket ocean" exhibiting many processes that are met pervasively and are of primary interest in the functioning of the global ocean (Robinson et al. 2001). It thus provides the opportunity to investigate 70 a large panel of oceanic features in a relatively restrained and nearby area. Therefore, the PROTEVS-MED cruises potentially caught a multitude of physical processes in the North Western Mediterranean Sea.

### 2.1 Thermohaline circulation

The basin or sub-basin scale circulation is largely dominated by the thermohaline circulation. The Mediterranean Sea is a 75 semi enclosed evaporation basin sea including areas of intermediate to deep convection. Less dense (and fresher) Atlantic Water (AW), inflowing through the Gibraltar Strait circulates roughly in cyclonic way in both Western and Levantine basins (Millot and Taupier Letage, 2005). This slope current is unstable along the Algerian Coast and generates anticyclonic eddies called Algerian Eddies (AE) that spread AW in the southern half of the Western Basin called Algerian Basin (Escudier et al., 2016a; Puillat et al.; 2002). In the Northern part of the Western Basin, the AW composed East Corsica Current (ECC) and 80 West Corsica Current (WCC) join in the Ligurian sea to form the North Current (NC) that flows along the slope until the Balearic Sea (Millot et al., 1999; Send et al., 1999). The existence and the strength of a return branch of this current along the North Balearic Front (NBF) between Menorca Island and Corsica Island is still under debate despite the generally accepted concept of a Northern (cyclonic) Gyre, in agreement with the doming of isopycnals in the central part of this sub-basin. The Levantine Intermediate Water (LIW) which is a modal water formed in winter in the eastern basin and entering 85 into the western basin through the Sicilian Strait follows more or less the same cyclonic circulation pattern. It spreads out into the northern part of the western basin between 400 and 800 m depths and is found sporadically within the Algerian Basin. It is spotted by a relative subsurface maximum of temperature and salinity.

### 2.2 Mesoscale structures

As in the global ocean the mesoscale dynamics is ubiquitous within the Mediterranean basin; it plays a major role in redistributing water masses and has been evidenced by remote sensing for a long time (Millot et al., 1990). In the Western Basin, the first internal radius of deformation spans in the range of 6 km in the Northern Gyre and of 16 km in the Algerian Basin (Escudier et al., 2016b). It is an indicator for the typical size of the mesoscale activity because surface intensified eddy sizes range in few deformation radii. As a result of ocean-atmosphere exchanges, large structures instabilities or flow-95 topographies interactions, Submesoscale Coherent Vortices, hereafter named SCV (Mc Williams,1985), with sizes currently



close to the local radius of deformation, have been observed (Bosse et al., 2015). Eddies, meanders, filaments and fronts are typically smaller and more contrasted than in the world ocean.

### 2.3 Sub mesoscale structures.

There are strong interactions between mesoscale structures thus generating intense stirring, layered structures and patchy ocean. Air-Sea exchanges are marked by a succession of strong events (Tramontane and Mistral gusts for instance in the western Basin); they interact with the mesoscale structures in generating sinks or sources of potential vorticity (PV), thus leading to ageostrophic dynamics.

Besides, the north western mediterranean basin is known to be the place of deep convection events which has been studied
for a long time and even taken as one of the paradigms of deep oceanic convection (Marshall and Schott 1999). Both modelling (Jones and Marshall, 1993, 1997) and observations (Bosse et al., 2016, Margirier et al., 2017) data shows that deep convection is highly favourable to the production of fine scale structures at submesoscale whether they are due to deepening of fronts during winter or to postconvection restratification.

### 2.4 Previous high-resolution observations

The finest part of the mesoscale dynamics often escapes the usual sampling strategy (CTD arrays, glider deployments) because being short lived, small in size and quickly advected. The development in the last decade of gliders fleet revealed nevertheless the mesoscale variability in the Western Mediterranean Basin.

Recent field experiments based on the multi-platform integrated monitoring program MOOSE (Houpert et al., 2016) or on
intensive targeted experiment HYMEX (Estournel et al., 2016) have revisited the hydrography and the dynamics of the North-Western part of the Western Basin. A strategy of regular and repeated gliders routes as well as dedicated deployments allowed to characterize the variability of the dynamics and to describe opportunely sampled fine scale structures. Bosse et al. (2015; 2016) inventoried the Submesoscale Coherent Vortices (SCV) and their contributions to water mass redistribution. With data issued from the same strategy, Margirier et al. (2017) characterized the convection plumes in the Gulf of Lions.
Testor et al. (2018) summarized the observations of convection during the dedicated experiment HYMEX. Multi-platform strategies including gliders, mooring, combined cruises ( Onken et al., 2018; Pascual et al., 2017; Petrenko et al., 2017; Knoll et al., 2017; Ruiz et al., 2009; Troupin et al., 2019), or colocation with altimetric tracks (Borrione et al., 2016; Heslop et al., 2017; Aulicino et al., 2018, Carret et al 2019) can provide part of the missing synoptic view.

The capability of changing the glider's trajectory at any time has not often been used in a small-scale context because its horizontal velocity remains low (in the range of 24 nautical miles per day), preventing any rapid assessment of a detected

small structure. Despite this lack of synopticity, Cotroneo et al. (2015; 2019) adapted a glider trajectory to a remote sensed observed Algerian Eddy and Current. Conversely the SeaSoar horizontal velocity is 10 times faster than the glider one ; besides, this towed vehicle can handle a turning radius of 2 nautical miles (i.e. gyration speed of 10 degree/min). It allows a
strategy based on long exploratory transects as the ship velocity is close to its transit velocity and on intensive sampling on particular detected structures. Due to heavy logistics involvement, the use of SeaSoar remained scarce in the Western Mediterranean Sea. Allen et al. (2001; 2008) observed an oblate lens of 20 km radius, 150 m thick, centred at 250 m depth during the OMEGA-2 field experiment in fall 1996. Salat et al. (2013) reported Seasoar transects in the Gulf of Lions after the convection in spring 2009. The Seasoar was also used during one leg of the ELISA field experiment devoted to the
Algerian Eddies (Taupier-Letage et al., 2003) but only the mesoscale features have been reported.

A free fall recovered vector, the Moving Vessel Profiler (MVP-200) has a lighter logistics but requires a lower vessel velocity to reach depths equivalent to those reached with the SeaSoar: that is to say 2-4 knots to go down 400 meter depths. In the Western Mediterranean Sea, the MVP was deployed during OSCAHR cruise allowing a detailed study of a cyclonic structure in the Ligurian Sea (Rousselet et al., 2019) and in situ estimation of the sea surface height for a comparison with
along track satellite data (Meloni et al., 2019).

## 3 Objectives and achievement of the field experiments.

The main scientific objectives of the cruises were threefold:
- to assess the large-scale circulation features of the Western Mediterranean Basin, evaluating the water masses and the fluxes at different key points in the basin (The North Current, The East and West Corsica currents, the Balearic front, the Algerian Basin). The final goal was some assessments of numerical simulation.

- to identify and follow peculiar mesoscale structures such as surface eddies, modal weddies, submesoscale coherent vortices (SCV) meanders or filaments and explore their signatures on the sea surface height (altimetry) and their acoustic impact (i.e.
through their modulation of the sound propagation speed).

- to observe and interpret the submesoscale dynamics such as ageostrophic stirring, symmetric instabilities, mixed layer instabilities, subduction and convection.

Clearly, the main part of the present dataset is not devoted to track any climatic change in water mass properties; the
SeaSoar, the MVP or the Rapidcast are rapid moving vectors leading to acquire less precise temperature, conductivity and above all deduced salinity data than standardized CTD's protocol. MVP and RapidCAST are equipped with unpumped sensors and the three reach high ascending or descending velocity (above 2m/s) that lead to inescapable thermal lag issues across sharp fronts. Readers interested in this topic should only use the CTD casts data.



Four cruises were conducted between 2015 to 2018 by the "Service Hydrographique et Océanographique de la Marine" (Shom) in the Western Mediterranean Basin, during winter or early spring, managing mainly the towed undulating vehicle SeaSoar to investigate the surface (0-400m) layer. When the deployment of this vehicle was either unsafe (over shallow water) or even impossible (due to rough meteorological conditions, breakdown of winch or vehicle) or when complementary observations were requested (e.g. go below 400m or getting water samples for biochemical analysis), CTD casts were rather

performed. Routinely acquired data (Vessel Mounted Acoustic Doppler Current Profiler (VMADCP) and ThermoSalinoGraph (TSG)) were also included in this database.

We present here in a synthetic dataset all the data recorded during the cruises (figure 1 ; table 1). Complementary data used to the cruise design, to adapt on field the strategy or to interpret results (altimetric tracks, remote sensed sea surface

temperature or chlorophyll) are available on CMEMS servers (http://marine.copernicus.eu). An eddy detection tools called AMEDA (Le vu et al., 2017) has also been used to track structures.

| protevsmed | 2015_leg1 07-01 / 24-01 | 2015_leg2 16-04 / 03-05 | 2016 22-03 / 04-04 | 2017_leg1 27-01 / 07-02 | 2017_leg2 11-02 / 23-02 | swot_2018_leg1 23-04 / 26-03 | swot_2018_leg2 30-04 / 18-05 |
|---|---|---|---|---|---|---|---|
| **SEASOAR** | **2290 km 1137 profiles** | **329 km 263 profiles** | **2090 km 1369 profiles** | **1858 km 706 profiles** | **620 km 1162 profiles** | **615 km 411 profiles** | **2830 km 2381 profiles** |
| **MVP** | | | | **153 km 813 profiles** | **188 km 708 profiles** | | |
| **RAPIDCAST** | | | | | | **22 km 92 profiles** | **167 km 71 profiles** |
| **CTD** | **62 profiles** | **151 profiles** | **47 profiles** | **17 profiles** | **27 profiles** | **1 profile** | **12 profiles** |
| **LADCP** | **56 profiles** | **137 profiles** | **47 profiles** | **18 profiles** | **26 profiles** | | **3 profiles** |
| **XBT** | **30 profiles** | **22 profiles** | | **39 profiles** | **1 profile** | | |
| **VMADCP 38kHz** | **2676 km** | | | | | **1071 km** | **4118 km** |
| **VMADCP 150kHz** | **3036 km** | **3688 km** | | **2705 km** | **5384 km** | **1068 km** | **4128 km** |
| **TSG** | **3756 km** | **3861 km** | **3744 km** | **2999 km** | **3011 km** | **1212 km** | **4984 km** |

**Table 1: Summary of performed transects, profiles and routinely acquired data. Cumulated length of transects and total numbers of vertical profiles are displayed.**




The first cruise called **Protevsmed_2015_leg1** took place from 7 to 24 January 2015 on board of the RV *Pourquoi Pas?*. The main objective was the dynamics of the North Current from the Ligurian Sea to the Gulf of Lions and the associated meso- and submeso-scale processes. Attention has been paid to transects across the slope of the Gulf of Lions, in order to examine the behaviour of the North Current and the exchanges across the shelf break. An intensive survey of the North

Current between Toulon and Nice was completed by drifting buoys deployment.

The second leg, **Protevsmed_2015_leg2**, was carried out on the RV *Beautemps-Beaupré* from 16 April to 3 May 2015. It started in the Balearic sea and investigated the slope current from Ligurian Sea to Balearic Sea. During this cruise, the SeaSoar trawl failed early, just after three transect acquisitions in the Balearic Sea describing the hydrology relative to the

cyclonic circulation and its associated mesoscale structures. This has led therefore to carry out mostly CTD casts and VMADCP 150 kHz records. In particular, a dense array of CTD casts was then performed within the North Current between Nice and Toulon. The protevsmed_2015_leg2 survey was characterized by a proliferation of jellyfish, the CTD measurements are to be taken with caution: a jellyfish on the sensors results in excessive smoothing of temperature and salinity. When too great differences appeared between the values at the ascent and descent, the profiles have been flagged 4

(bad value that can be corrected)

The second campaign, **Protevsmed_2016**, took place on the RV *Beautemps-Beaupré* from 22 March to 4 April 2016. It was designed to focus on the origin of the Northern Current which is known to be fuelled by the Corsica Channel flow and the Western Corsica Current. The latter is connected in a more or less clear manner to the North Balearic Front (NBF) and the

continental slope current. Besides unveiling part of the complex hydrological structure of the NBF at early spring, the Protevsmed_2016 survey allowed to catch an algerian eddy in interaction with the NBF. Garreau et al. (2018) described in details its original double core structure: a superposition of two water masses of different origin spinning together. The survey provides also scenes and insights on the way both components of the North Current merge together to the north of the Corsica Channel during the early spring.


The third campaign **Protevsmed_2017** held from 27 January to 7 February (**leg1**) and from 11 February to 23 February (**leg2**) on board the R/V *Atalante*. This survey was devoted to explore eddies detected by altimetry in the North Balearic Front to assess an eddy tracked tool (Le Vu et al., 2017). Transects across the North Balearic Front revealed the complexity of this transition zone. In order to escape rough sea state sequential to a strong Mistral gust part of the cruise was dedicated

to the investigation of the Balearic Sea and the outflow of fresher and colder water from the Gulf of Lions. Back to the deep-sea area, partial convection and Western Intermediate Water (WIW) formation of water were recorded. An SCV were thoroughly observed, north of the Balearic front.



The fourth and last field experiment, **Protevsmed_swot_2018** was conducted in the framework of SWOT Preparatory Phase
from 23 April to 26 April (**leg1**) and from 30 April to 18 May (**leg2**) south from Balearic Islands on board the R/V
*Beautemps-Beaupré*. The first leg (**leg1**) performed a general overview of the oceanic situation followed by a more intensive
survey (**leg2**) planned on the basis of daily releases of near real-time satellite imagery, altimetry, and Lagrangian analyses,
performed on land by use of the SPASSO package (http://www.mio.univ-amu.fr/SPASSO/ , as in Nencioli et al., 2011; de
Verneil et al., 2017**).** Satellite data of altimetry, sea surface temperature and ocean colour revealed ubiquity throughout the
cruise period of very fine oceanic structure such as mushrooms-like structures or tenuous front.  A special focus was stressed
on the South of Mallorca where a frontal zone has been detected by altimetry-derived currents and diagnosis (e.g. Finite
Singular Lyapunov Exponents), by contrasted surface chlorophyll concentrations and confirmed by high frequency flow
cytometry analyses of phytoplankton performed onboard (data not included in the present dataset). A Lagrangian round-trip
strategy was specifically set up in order to study the structure and growth rate (at 24 hours time scale) of the various
phytoplankton groups as defined by flow cytometry measurements as in Marrec et al. (2018). Last, it is noticeable that a
companion campaign (PRE-SWOT) managed by IMEDEA-SOCIB held in the same area and during the same period on
board of RV *Garcia Del Cid* (not included in the present dataset, see Barceló-Lull et al., 2018).

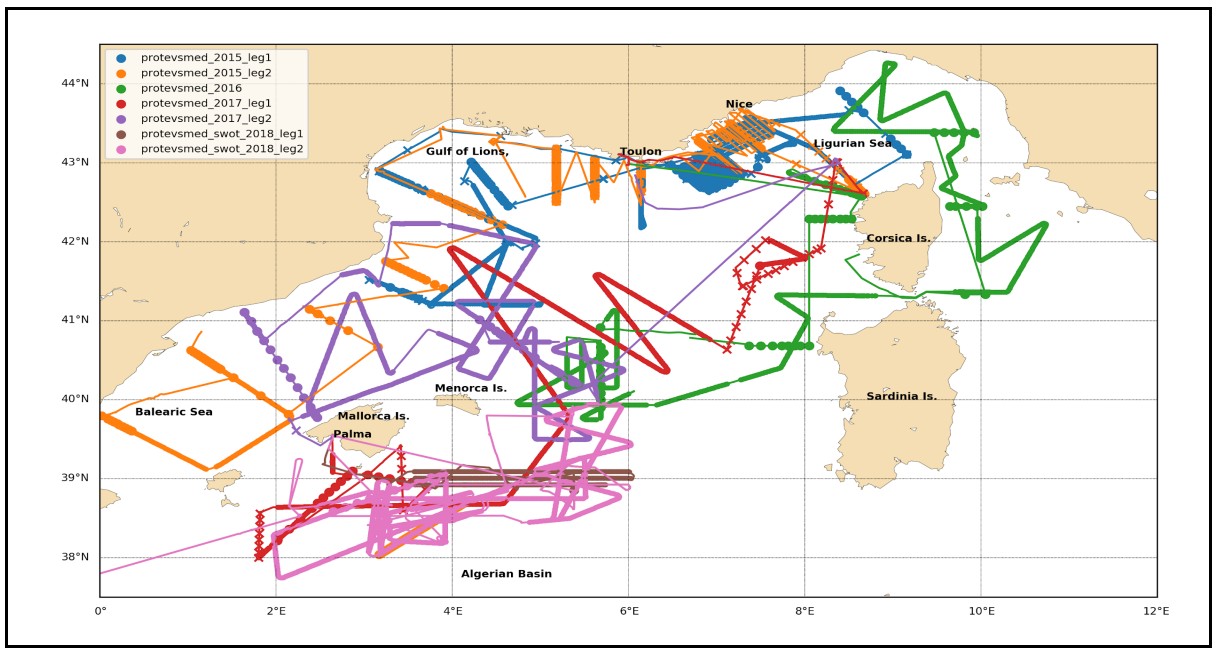

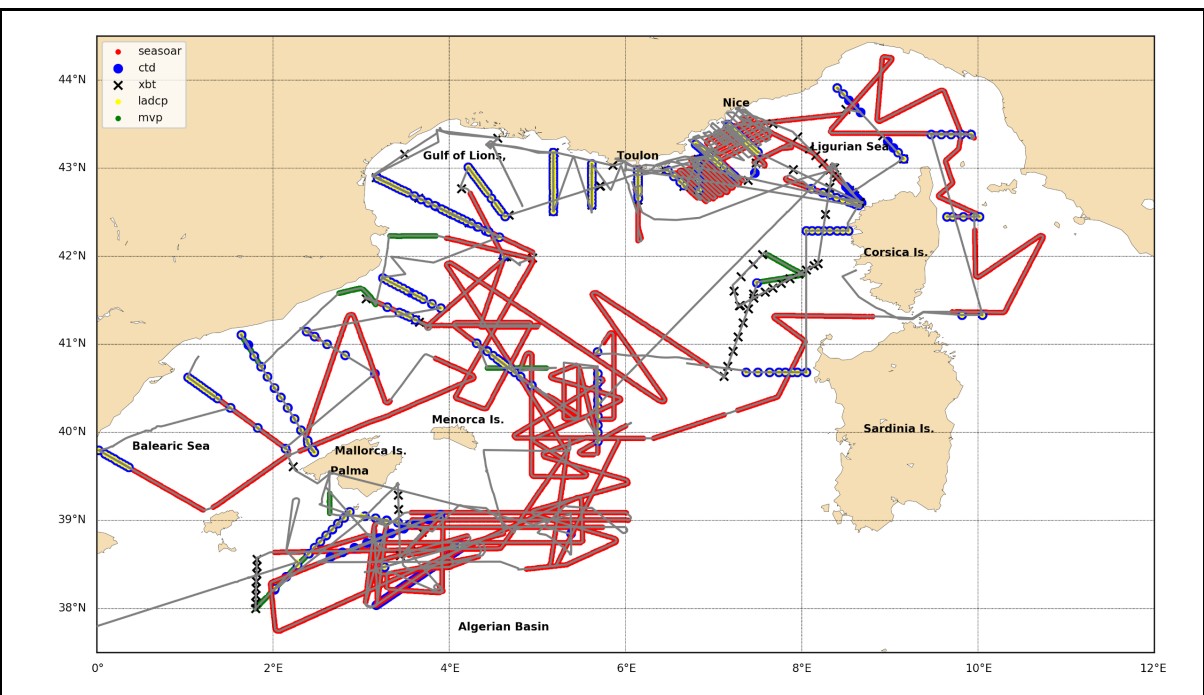

**Figure 1: Map of Protevsmed cruises (a) and instruments deployed (b); TSG is available for all the ship track.**


# 4 Data, Methods and Quality Controls

## 4.1 CTD casts and LADCP

The CTD casts were performed with the Sea-Bird SBE-9 instrument mounted in a General Oceanics 12-places rosette frame
fitted with 12 Niskin bottles. Sometimes an RDI 150 kHz current profiler was also implemented on the rosette and then
LADCP (Lower Acoustic Doppler Current Profiler) performed measurements during the cast. Standard hydrographic
procedures for CTD casts were applied. When available, LADCP recorded data were processed using the software developed
by Visbeck (2002).

## 4.2 Seasoar deployments

The SeaSoar is a towed undulating vehicle designed and built by Chelsea Instruments. Two Sea-Bird SBE-9 (with SBE-3
temperature and SBE-4 conductivity sensors) instruments were mounted on either side of the SeaSoar. When available, a

chlorophyll *a* WetStar WET Labs fluorometer, both oxygen sensor (SBE-43) and optical properties sensor (WET Labs C-Star) were deployed. The SeaSoar was trawled at 9 knots linked to the board by a profiled cable. It was undulating between

the surface and 400m below the surface under optimal conditions with a horizontal resolution in the range of 1 nautical mile. Rough sea states, lateral currents, strong vertical shear can degrade the performance of the vehicle and reduce the vertical range of exploration between 20 and 360m. As the software allows on time visualisation of the ongoing transect, it is a perfect tool to scan the upper oceanic layer where meso and sub-mesoscale dynamics is the most intense A total of 5400 nautical miles of transects crossing plenty of different structures has been yet recorded during the four cruises, given the

unique opportunity to explore the fine scales patterns of the upper layer of the western basin.

### 4.3 MVP deployments

During the Protevsmed_2017 surveys a Moving Vessel Profiler (MVP200) - a computer controlled winching system that can deploy and recover a sensor from a ship that is underway - was deployed for finer transects. The Sensor was an AML CTD

embedded in a free fall fish. At 2-4 knots, it was possible to monitor the 0-400m layer with a horizontal resolution of a half nautical mile. To remove spurious salinity values due to bubbles when fish is surfacing, the minimum pressure for valid record was set to 1 decibar. A peculiar transect has been monitored using successively SEASOAR, MVP and CTD casts given the opportunity to compare the three techniques.

### 4.4 RapidCast deployments

During Protevs2018 a free fall CTD system, called rapidCast (Teledyne Marine ; http://www.teledynemarine.com/rapidcast) was tested for three transects near Balearic Islands. It was equipped with the "rapidCTD - Underway Profiler" proposed by Valeport. A bluetooth communication allowed the real-time evaluation of each profile when the probe is surfacing near the ship deck. This system sampled the water layer from 0 to 400m with a navigation speed in the range of 5-6 knots.


### 4.5 TSG

During the cruises, a Seabird SBE-21 ThermoSalinoGraph recorded the sea surface temperature and conductivity. The in duct was equipped with a SBE 38 thermometer. The recorded sea surface temperature and salinity contribute to the Global Ocean Surface Underway Data (GOSUD) program. The metrological traceability and the data treatment are insured

according to procedures described in Gaillard et al. (2015) which explains the delayed mode processing of datasets and presents an overview of the resulting quality. The calibrations are complemented with rigorous adjustments on water samples leading to reach a salinity accuracy of about 0.01 or less.



**4.6 VMADCP**

The hardwares used, their configurations and the way they are carried out are similar on both R/V Pourquoi Pas ?, R/V L'Atalante and Beautemps Beaupré. The VMADCPs are of the type Ocean Surveyor by RDI Teledyne : the R/V hull is equipped with two antennas one at 150 kHz and the other 38kHz. They are both monobloc antennas using beam forming process to form four beams oriented towards 30° from the vertical. Nominally, they emitted a ping per second from which ensemble are built to get a less noisy profiles. Two ensembles are routinely processed:

-    a Short-Term Average (hereafter noted STA), which gather and averaged the pings of two-minute window. It makes ensemble of 120 pings at least,

-    a Long-Term Average (noted LTA) made of 600 pings or averaged over ten minutes.

The series of geometric transformations necessary to pass from beam coordinate along beam data to absolute geographic coordinate and geophysical velocity are performed thanks to VMDAS software from RDI Teledyne. It combines the position

(latitude longitude) from the DGPS Aquarius and Octans central with the PHINS inertial navigation system form IXSEA (that provides vessels attitude data: pitch, roll, heaving) to provide synchronised single ping earth coordinates data (file .ENX) and short- and long-term ensemble (STA / LTA).

This native format (.STA/.LTA) are also processed with WinAdcp in order to extract and provide only significant (i.e. with a satisfactory signal/noise ratio) data that are additionally formatted to text file.


Note that data processed by CASCADE software can be requested on Sismer repository, collecting and processing progressively all VMADCP from French research vessels ([https://sextant.ifremer.fr/record/60ad1de2-c3e1-4d33-9468-c7f28d200305/en/index.htm](https://sextant.ifremer.fr/record/60ad1de2-c3e1-4d33-9468-c7f28d200305/en/index.htm)).

**4.7 Data metrological traceability and calibration**

SBE 9 temperature and conductivity sensors deployed on CTD and Seasoar were calibrated before and after each campaign or at least once a year in the Shom's thermo-regulated bath; its temperature can be stabilized to less than 1 mK (peak to peak) during control and calibration operation. Such a procedure allows the monitoring of sensors drifts between calibrations and the detection of anomalies. In the cases sensors have kept a good linearity, which is the more common, data are

corrected with offset-slope coefficients. Figure 2a shows the review of corrections applied on data at 15 °C, after the calibrations of SBE 3 sensors used for PROTEVS-MED campaigns. Figure 2b shows the review of corrections applied at 40 mS cm$^{-1}$ after the calibrations of SBE 4 sensors.





The temperature of the thermo-regulated bath is monitored with SBE 35's which are used as laboratory reference temperature sensors. They are linked to the International Temperature Scale of 1990 (ITS-90) thanks to calibrations performed once a year in a triple point of water cell and in a melting point of Gallium. These reference cells are regularly calibrated by the French National Metrology Institute (NMI) LNE-CNAM. The calibration expanded uncertainty of SBE 3 sensors is between 1.8 and 2.3 mK according to the residual linearity errors of SBE 3's.

Conductivity calibration of SBE 4 sensors is made in the same bath during the temperature calibration. Seawater samples are taken in the bath and tested with one Autosal and one Portasal salinometer. The calibration procedure and the propagation of uncertainties to the calculated salinities from SBE 9 data are described in Le Menn (2011). Practical salinity expanded uncertainty varies from 0.0032 to 0.0034. In 2015, the Shom laboratory took part in the JCOMM intercomparison for seawater salinity measurements (JCOMM, 2015) showing that Autosal and Portasal measurements are within ± 0.001 compared to other participating laboratories. Note that the same process was done, in the framework of an international network, for the TSG data of the French Research Vessel (see §4.5 and table 2). Unfortunately, the MVP and the RapidCast sensors were not available for such a common process and were calibrated directly by the provider. As the optical properties and oxygen concentration were used as tracer only, no calibration process was performed.

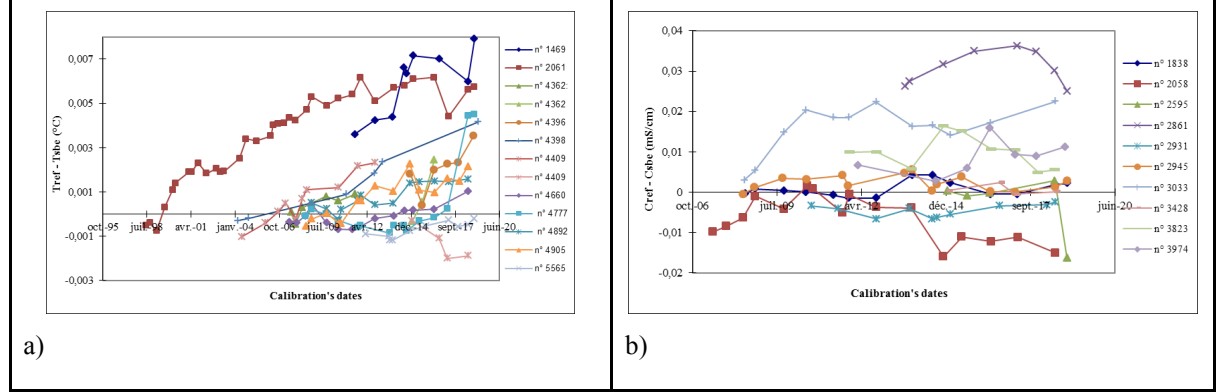

**Figure 2 (a) review of corrections applied on data at 15 °C, after the calibrations of SBE 3 sensors. (b) review of corrections applied on data at 40 mS cm$^{-1}$, after the calibrations of SBE 4 sensors.**

### 4.8 Data processing levels

Three levels of processing are available for data:

- The first one (Level 0, L0) consists in direct output of sensor at full temporal resolution;
- The second level (Level 1, L1) displaying data in ascii (.csv) or netcdf (.nc) files are only processed from the software of the constructor, keeping the full resolution and computing the derived variables into standard units.



Recent instrumental system (AML and Valport probes) directly provides level 1 files. L1 files are corrected from eventual drift of sensors;

- The third level (Level 2, L2) is proposed as gridded, controlled and resampled data in netcdf files (.nc). Gridded dataset for salinity and temperature have been resampled every meter removing spike, spurious values, density inversion when they persist after the first process supplied by the sensor manufacturer.

Temperature and salinity data were also compared to the historical data in the neighbourhood of the profiles or transects 330 using the validated CORA database distributed by the Copernicus Marine and Environment Service (Cabanne et al., 2013; Szekely et al., 2017). Protevsmed data are not yet included in this database but will be transmitted for a future release.

All gridded profiles or transects have been plotted for a visual quality check and are available as "quick looks" on the repository. Level 1 (L1) and (or) level 2 (L2) dataset are released in the present database. Level 0 (L0) remain available in constructor format upon request to the data providing institution (SHOM).


### 4.9 Companion datasets.

During the field experiments, surface drifters with holey socks located at 75m, 50m or 15 m depth were deployed given the opportunity at the beginning of their track to perform a lagrangian survey of observed structures.

Few Argo floats were also dropped and experienced Protevsmed dedicated profile (higher temporal resolution, parking depth into the targeted structure, etc.) at the beginning of their mission before shifting to the usual Argo standard procedure in Mediterranean Sea (i.e. 5 days cycle and parking depth set to 350m). Already stored in dedicated and accessible stable repositories, they can be found using their WMO identifiers (World Meteorological Organisation) (table 2). Ancillary data can be found on different repositories selecting date and location corresponding to Protevsmed surveys.


| | ARGO float WMO identifier<br><br>http://www.ifremer.fr/argoMonitoring/ | SVP surface drifter WMO identification number (holey sock depth)<br><br>http://www.jcommops.org/dbcp/ | TSG ThermoSalinoGraph (platform identifier)<br><br>http://www.gosud.org |
|---|---|---|---|
| protevsmed_2015_leg1 | **6901707**<br>**6901708** | **6100536 (50m)**<br>**6100537 (50m)**<br>**6100538 (50m)**<br>**6100539 (50m)**<br>**6100540 (50m)** | **FMCY** |
| protevsmed_2015_leg2 | | **6100863 (50m)**<br>**6100864 (50m)**<br>**6100865 (50m)** | **FABB** |





| | | 6100866 (50m)<br>6100867 (50m) | |
|---|---|---|---|
| protevsmed_2016 | | 6101525 (75m)<br>6101526 (75m)<br>6101527 (75m)<br>6101528 (75m)<br>6101529 (50m)<br>6101530 (50m)<br>6101531 (50m)<br>6101532 (50m)<br>6101533 (50m)<br>6101534 (50m)<br>6101535 (50m)<br>6101536 (50m)<br>6101537 (50m)<br>6101538 (50m)<br>6101539 (50m)<br>6101540 (50m)<br>6101541 (50m)<br>6101542 (50m)<br>6101631 (50m) | FABB |
| protevsmed_2017_leg1 | 6902764<br>6902765<br>6902767 | 6101634 (50m)<br>6101639 (50m)<br>6101635 (50m)<br>6101637 (50m)<br>6101636 (50m)<br>6101643 (50m)<br>6101641 (50m)<br>6101647 (50m) | FNCM |
| protevsmed_2017_leg2 | | 6101648 (50m)<br>6101633 (50m)<br>6101638 (50m)<br>6101640 (50m)<br>6101632 (50m)<br>6101642 (50m) | FNCM |
| protevsmed_swot_2018_leg1 | 6902844 | 6101669 (50m)<br>6102612 (50m)<br>6101677 (100m)<br>6102613 (100m) | FABB |
| protevsmed_swot_2018_leg2 | | 6101671 15m<br>6101678 15m<br>6101672 15m<br>6102615 15m<br>6101670 50m<br>6101674 50m | FABB |

**table 2 : WMO index of Argo-float and surface drifters (SVP) dropped during Protevsmed surveys. Surface temperature and salinity recorded by ThermoSalinoGraph (TSG) are tagged by ship identifier. All data are available from the CORIOLIS website http://www.coriolis.eu.org/Data-Products/Data-Delivery/Data-selection by entering the WMO numbers in the field 'Platform codes', adjusting the time period of interest (e.g., 01/01/2018 to 30/06/2019), and clicking on 'refresh'. The web interface displays the trajectories of the buoys, profilers or TSG and can be used to find additionally opportunity data. The data can then be downloaded in NetCDF format.**




## 5 Overview of the observations


Deployed together, VMADCP and SeaSoar provide a unique synoptic view of a transect. It is also possible to observe simultaneously the density and the velocity fields in the sub-surface layer, showing the predominance of the geostrophy even at fine scale (few kilometres). The temperature and salinity fields are patchier than expected but the thermal expansion and the saline contraction coefficient of sea water often compensate and lead to a smoother density (and thus dynamical) field.

The structure of observed anticyclonic eddies appears also more complex than formerly described and are currently composed of many different water masses. Eddies with similar altimetric or surface thermal signatures can hydrographically be very different. For instance, a dual core anticyclonic eddy has been observed east of Menorca Island in March 2016 (figure 3a), composed of a superposition of Winter Intermediate Water and Atlantic Water and, in May 2018, a three levels eddy has been detected (figure 3b). The exact process of their formation remains still debated. One can invoke the

coalescence of pre-existing objects, the extraction of water masses from neighbouring structures or ageostrophic processes. Measurements reveal submesoscale (ageostrophic) dynamics both in the eddy cores (upwelling/downwelling) and at eddy edges (symmetric instabilities). Intra-pycnocline structures subducted, stirred or locally formed were commonly observed at the edge of gyres.

In the North Current, stirring appeared in both tracers and velocity fields and an SCV formed by LIW detached in front of

Toulon (figure 3c) was observed. Note that the anticyclonic SCV was topped by a surface cyclonic gyre as confirmed by VMADCP record (not shown here). Current were routinely recorded and a particular attention has been paid to the North Current dynamics (figure 3d).

The fine structure of the NBF showed the interaction between the front and the SCV's generated in the Northern part of the Basin. In the NBF, a shift between a surface layer front and a deep front is revealed. As long as most of the experiments

presented here were performed during late winter or early spring in the vicinity of the north western convection area, they frequently shows small scale structures that are likely post convective. For instance, from the North to the South on figure 3e one can observe successively the - probably partial - convection area, the formation of SCVs composed of WIW in the mesoscale adjustment area around the chimney, the surface thermal front and finally a deeper front at 40.6° N.

Over the abyssal plain next to the bottom, CTD casts highlighted different Western Mediterranean Deep Waters (WMDW) ;

besides, under the LIW layer, where the profiles foster double diffusive process, staircases in temperature and salinity were commonly observed (figure 3f).

MVP transects performed in February 2017, in front of cap Creus, across the Blanes Canyon, and off Barcelona showed cold water diving along the Catalan shelf and slope (figure 3g). The WIW existing along the Catalan slope was relatively fresh and cold water in-flowing from the Gulf of Lions shelf. The WIW took progressively place between the Atlantic Waters

(AW) and the LIW as they flowed southwards.

Similar and extensive quick looks of all Seasoar, MVP, or Rapidcast transects and XBT, CTD profiles are plotted and available as additional resources on the data repository.



**Figure 3. Overview of some transects or profiles recorded during the Protevsmed fields experiments: (a) Dual core eddy in NBF in 2016, (b) Three layered eddy in Algerian Bassin (2018), (c) SCV of LIW front of Toulon (2015), (d) North Current position and intensity (2015), (e) cross frontal (NBF) transect (2017), (f) staircase in temperature front of Sardinia (2016), vein of cold water from the Gulf of Lion on Catalan sea (2017). The reader will find similar quick looks of the transects for all the surveys into the data repository.**

## 6 Conclusions

The PROTEVS-MED dataset available through an unrestricted unique repository is an unprecedented opportunity for the community to approach the very fine scale dynamics in the Western Mediterranean Sea and more largely the sub-mesoscale

dynamics associated with strong mesoscale dynamics. In the framework of the high-resolution altimetry this dataset can help to characterize the scales of fine structures in the Western Mediterranean Sea and to design combined experiments using

high resolution In-Situ measurements (Seasoar or MVP) and altimetry with the future SWOT satellite (d'Ovidio et al., 2019). It should be complementary data set to usual glider one and useful to design future combined surveys.

During these campaigns, we had the opportunity to deploy different devices to obtain temperature, salinity and possibly other parameters profiles. Some transects has been performed successively using CTD and Seasoar (all surveys) or using

CTD, Seasoar and MVP-200 (Protevsmed_2017). Easy to manage, the Rapidcast has been only used for test and as instant spare in 2018 when the Seasoar failed. It produced a similar result to the Seasoar in temperature and salinity. The SeaSoar is heavy to manage, need a consequent research vessel for the winch system, a constant watch on its navigation and calm sea state for water launching and recovery. Once deployed, the machine can stay at sea for days. Thanks to the required ship velocity (about 9 knots), the SeaSoar remains a perfect machine to drive out mesoscale structures before examining them in

detail. It explores the oceanic surface layer up to 400m deep which is sometimes a little bit too short in Mediterranean context, missing deeper part of AEs or deep SCVs but sufficient to describes the main part of the dynamics. For the same depth range a MVP-200 requires a ship velocity about 2-4 knots and is then more devoted to short transect with higher horizontal resolution. As it is a free-fall vector, it setting up is lighter, despite regular inspection of the cable and winch every 10 hours. In any case, when exploring a structure in detail, a CTD network remains necessary, at least to have a valid

reference level for the thermal wind equation.

Despite the suspected lack of accuracy due to the velocity of the vectors (Seasoar, MVP, RapidCast), it is demonstrated in these datasets that fast and high-resolution sampling reveals fine oceanic patterns never described before in the Western Mediterranean. In situ observations of ageostrophic dynamics remain scarce and the back and forth between these

observations and theory, then between these observations and modelling should be very fruitful. These data should contribute to the knowledge of small scales and fill some of the gaps in observing system in the Mediterranean Sea (Tintore et al., 2019). As numerical modelling gain in resolution (until a few hundred meters), the simulation of sub-mesoscale processes (layering, subduction, stirring, vertical velocities) is therefore expected and this dataset, providing data at similar scale, is an opportunity to validate the secondary simulated circulation.


**Authors contributions**

Louazel S., Correard S., Garreau P., and Dumas F. designed and conducted the field experiments as PI. Marc Le Menn

managed the calibration and the metrological traceability of SBE sensors. Valerie Garnier has carefully checked the dataset.

All co-authors carried them out, participated to the cruise or processed the data. Garreau P. and Dumas F. prepared the manuscript and the data with contributions from all co-authors.

**Competing interests.**

The authors declare that they have no conflict of interest.

**Acknowledgements**

The authors acknowledge that the French Government Defense procurement and technology agency (Délégation Générale de l'Armement) funded extensively through Protevs and Protevs II "programme d'étude amont" all the campains reported here.

They thank the technical team at the French Naval Hydrologic and Oceanographic Service (SHOM), the crews of the French Navy ship *Beautemps-Beaupré,* of the RV *Pourquoi Pas ?* and the RV L'*Atalante* for their contribution to the field experiments.


**Data Availability and Repository**

Data are freely available on SEANOE repository (https://doi.org/10.17882/62352; Dumas 2018). Some parts of the data are already under investigations or publications; the authors would appreciate collaboration proposals. For a first overview quick

looks of all Seasoar, MVP, or Rapidcast transects and XBT, CTD profiles are available in catalogues on the repository.

Seasoar, MVP, RapidCast, CTD, LADCP and XBT data are stored in both CSV (ASCII) and Netcdf files for "L1" (directly extracted from the instrument or constructor software), in Netcdf for "L2" (resampled every meter) files;

 For TSG, the present database provides only L1 files; L2 (validated and resampled data) are available on dedicated repository (see table 2).

For the sake of simplicity, VMADCP files were concatenated over each cruise duration to provide a single file per cruise; for a given cruise, the data are a function of time and depth within the single file dedicated to the cruise

Data are displayed by cruises and instruments and the syntax is :

**intrument_data-type_cruises_starting-date-of-record_index.file-type** , where



**instruments** = ctd,seasoar,ladcp,xbt,rapidcast,mvp,vmadcp_xxx

**data-type** = "L1" or "L2"

**cruises** = cruise and leg name

**date** =  the date of the first record in the file.

**index** = sequential index of this kind of profile recorded during the cruise.

**file-type** = csv(.csv) or netcdf(.nc)

Additionally, data extracted from on-board automatic acquisition are provided in Netcdf file for the ship navigation. Future PROTEVS_MED experiments are scheduled and results will be added to the repository.

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
