# Peer review of "PROTEVS-MED field experiments: Very High-Resolution Hydrographic Surveys in the Western Mediterranean Sea"

_Earth System Science Data, 2019_

## Referee Comment (RC1) · Anonymous Referee #1 · 17 Oct 2019

In this manuscript, the authors present data from a series of field experiments realized in the Western Mediterranean Sea from 2015 to 2018. Data collection is mainly realized in two seasons through the use of several platforms as towed vehicles, free falling profiler, classical CTD stations and current proflers. The resulting dataset describes the physical properties of the water column at different spatial resolution according to the working depth of the instruments. It is worth mentioning the high level of resolution and ability to describe mesoscale and sub-mesoscale processes achieved in the very first layer of the water column. This resolution (about 1 nautical mile) is also typical of glider measurements, nevertheless, in this work, data presented have been collected in a shorter time, allowing a higher degree of synopticity to be achieved. Data collected along depth has been also completed by undergoing measurements collected at surface on board the research vessel and by biological data collected through dedicated sensors. The paper is easy to read, well-structured and provides detailed description of the data and of the field experiments designed to collect them. Datasets are easy to access and are of full interest for scientists focusing on mesoscale and sub-mesoscale processes, even if at least one link (https://www.seanoe.org/data/00512/62352/data/66880.pdf) is not working.

I consider that this paper must be published after minor revision. Here are some comments or questions and a few technical remarks that can be useful for the authors and that should be addressed.

Line 24 "(about 10000 Km)" could be replaced by (total length about 10000 Km)

Line 28 TermoSalinoGraph (TSG) "CTD casts" could be replaced by Classical full depth CTD stations have been realized.... Please define CTD acronym. I think that the manuscript would benefit of a clear distinction between a CTD (instrument that is included in the towed system as well as in the free fall profiler) and a classical CTD cast performed when the ship stops in a sampling station. At the moment, in this manuscript the term CTD is used for all the classical oceanographic stations and this could generate some confusion.

Line 29 "objects" may be replaced by "structures"

Line 30 "the aim of the survey...." Please consider resentencing

Line 32 "biological sensors....have been carried out" Please consider resentencing

Line 44 "chlorophyll a " replace with chlorophyll a. Please be coherent throughout the paper

Line 47 "As the scales" might be replasec by "As all these scales" "to develop observations" please replace with"to develop an observation strategy" Line 56-60 Please consider to move here the figure 1 also adding the geographical references mentioned

in the text (up to section 3). I would also move to the very first lines the name of the study area. Line 57 "depths under" maybe "below"

Paragraph 2.1 Please consider to add a figure showing the Mediterranean Sea and the oceanographic features described in this paragraph, as well as the location of the study area. Please add a description of the deep layer properties, or alternatively rename the paragraph to focus on surface and intermediate circulation and main water masses

Line 101 "patchy ocean" maybe "patchy ocean areas"

Line 106 show

Line 112 because of

Line 121 high resolution in situ data by glider have also been compared to the new generation salinity products by SMOS satellite as in "Aulicino, G.; Cotroneo, Y.; Olmedo, E.; Cesarano, C.; Fusco, G.; Budillon, G. In Situ and Satellite Sea Surface Salinity in the Algerian Basin Observed through ABACUS Glider Measurements and BEC SMOS Regional Products. Remote Sens. 2019, 11, 1361".

Line 129 What you mean with "turning radius"? Is the ability to change direction?

Line 146 The parenthesis includes both oceanographic features that are described in the dataset and basins. Probably listing just one of the categories would be better.

Line 148 "weddies" have not been defined before

Line 155 "rapidcast" not mentioned or described before please add a description at lines 127-140 as for SeaSoar or MVP

Line 158 "CTD casts" maybe "classical CTD stations"

Line 160 "Shom" replace with "SHOM"

Line 165-166 Consider removing the inner parenthesis. i.e. "– VMADCP –"

Line 192-195 NBF and NC have already been defined, please use the acronyms

Line 204 "...to a strong Mistral gust, part of the cruise...."

Line 212 "lagrangian"

Line 217 "Finite Singular Lyapunov Exponents" consider adding a reference

Line 244 "given" or "giving"?

Line 280 (latitude and longitude)

Line 292 "Shom" should be SHOM. Please check throughout the entire manuscript

Line 294 in the case....which is the more common"

Line 298 "SBE 35's" maybe "SBE 3's"?

Line 319 "available for data" maybe "available for each dataset"

Line 320 "consists in" maybe "consists in the" "sensor" should be sensors"

Line 325 Are the gridded profiles averaged along depth only? The term gridded may confuse the reader

Line 340 What you mean with "higher temporal resolution"? Please clarify. If the float is enveloped in a structure it would provide longer observations in time.

5. Overeiew of the observations As stated in the introduction, here a limited number of sample analysis of the collected data are offered to the reader. Please consider adding a sentence at the beginning of the chapter that clarify this.

Line 356 "When deployed"

Line 376 "frequently show"

Line 379 "CTD casts" should be "classical CTD stations"

Line 379-381 Please add the position of these casts on figure

Figure 3 Please consider splitting this figure in order to obtain an hgher definition for

each plot

Line 398-401 Please consider resentencing

Line 404 "Some transects have been"

---

## Referee Comment (RC2) · Anonymous Referee #2 · 4 Nov 2019

This paper describes a unique dataset from towed instruments aiming at resolving the sub-mesoscale during several cruises in key areas of the western Mediterranean. The originality of the approach consist in the higher degree of synopticity compared to autonomous gliders, or ship measurements. The dataset resolves the upper 400m of the water column and covers a broad range of processes (eddy, front, convection...), which makes it very relevant for the community.

In a general manner, the paper is well organized and provide most of the requirements to be published in ESSD. However, I have some minor comments, mostly related to style and references, to make it clearer, consistent and easier to read.

[Figure]

L25 missing ,

L29 "peculiar objects" prefer particular processes ; a MVP

L30-32 ... to access even higher resolution the ocean physics (temperature, salinity, currents). Biological sensors were opportunistically used to provide ...

Throughout the paper : opportunely -> opportunistically

L33 the fine-scale processes

L35 properly résolve

L43 connecting ... to ; energy cascades to small scales and reversely

L44 For instance in the northwestern Mediterranean ; chlorrophyll-a

L46 vortices

L50-55 partly fill these gaps between the large and finer scale dynamics. Since many years, remotely ... observe a large ... soon by the future Surface Water and Ocean ... mission (SWOT), expecting to provide ... small-scale processes ...

L59 in a context of ocean physics I would use km instead of nautical miles.

L63 looks at ; are described

L66 Oceanic context of the

L68 is sometimes referred as a "lab-ocean"

L70 "nearby" did you mean close to Land? Accessible?

2.1 General circulation : you aren't describing the thermohaline circulation very much...

L74 sub-basin circulation ; what to you mean about dominated? Please rephrase.

L75 basin or sea, not both ; Light (fresh) Atlantic Water

L76 generally circulates along the continental slope ... western and eastern basins

L77 The slope ... Algerian coast

L79 and throughout the paper, no caps for western basin etc

L77-79 These lines fit better to 2.2

L79 northern part

L80 Ligurian Sea ; Northern Current is more often found (please modify consistently throughout the paper).

L83 (sounds more logic in this order) The generally accepted concept of the Northern Gyre flowing cyclonically around the doming of isopycnals of the deep convection area of the northwestern Mediterranean (please cite a ref about the general circulation). The existence, position and strenght of the return flow of this gyre ...

L84 (LIW), a mode water ...basin entering the western ... , follows pattern (again please provide ref)

L87 This important water mass is marked by a relative ...

L93 indicator of ; because the scale of surface-intensified eddies in geostrophic balance ranges a few ...

L95-96 topography ; whose size is close to the local ; observed in the Western Mediterranean (also cite Testor and Gascard 2003, 2006; Bosse et al 2015, 2016)

L97 contrasted in terms of what? What's the point of reference? Subtropical regions might not be comparable, while polar regions would tend to exhibit similar characteristics with density compensated contrasts.

L99 Submesoscale (tends to be found in one word in the litterature, please correct throughout the paper)

L101 marked by frequent strong wind évents ... northwestern basin ... with the NC and

mesoscale structures and generate ... (No need for PV acronym if not used afterward) ... please the relevant litterature : Bosse 2015, Estournel et al 2016, Giordani et al 2017, Testor et al 2018

L104 a place ; for a long time (cite for instance MEDOC group et al 1979, Schott et al 1996, Houpert et al 2016, Testor 2018)

L106 both models ... and observations ... show

L108 due to the deepening of the mixed layer

L111-115 ... are only partially resolved by usual ... ; repeated glider lines (instead of glider fleet) ... in the Western Mediterranean, in particular as part of the multi-platform ... MOOSE (rather cite Coppola et al 2019). Intensive targeted ... the dynamics of the deep convection area in the northwestern Mediterranean Sea (Estournel et al 2016).

L116 glider lines

L117 opportunely

L121-122 please sort references in chronological order

L126 prefer 15-30 km/day than nautical miles

L130 sampling of

Proposition of section/subsection titles: 3 The PROTEVS-MED field experiments 3.1 Objectives 3.2 Cruises

L146 key regions of the basin ; North Balearic Front

L147 "assessments of numerical simulation" is not very clear... What kind of simulation? How?

L148 please avoid modal weddies which is not defined earlier ; surface and subsurface mode water eddies (including SCV)

[Figure]

L149 meanders and filaments

L151 observe and characterize

L155 prefer platform to vectors... Also in other occurrences in the paper

L158 across strong thermocline ; shipborne CTD casts.

L161 SHOM?

L164 ship CTD

L166 ))

L168 dataset of all

L171 to detect and track mesoscale structures during the cruises.

L178 paid to cross-slope transects

L180 please refer to table 2. Apparently surface drifters were deployed during every cruise. Since it is not only the case for the first one, it feels odd to mention it here and not for the others.

L183 Balearic Sea (please check capital letters for the rest for the rest of the paper).

L185 therefore led ; mostly carry out ship CTD casts

L188 with caution, as they caused an excessive ...

L193 origin of the NC where the flow through the Corsica Channel and the WCC join.

L194-195 This latter ... current. I am not sure about what the authors are trying to say here.

L195 in early spring

L196 to capture an Algerian Eddy

L198 also provides insights about ; Northern Current

L203 eddy tracking tool

L204 consecutive to a strong Mistral gust, ...

L206 (WIW) formation were ... ; A SCV was

L212 release

L213 what is the acronym of SPASSO?

L215 mushrooms-like structures -> do you mean dipolar structures ? ; fronts ; focus was given to the area south of Mallorca where

L216 a front was detected ; Lagrangian diagnosis

L218 "A Lagrangian round-trip strategy" I dont know what this is...

L230 a RDI 150khz ... Was the LADCP profiles acquired by a pair of those instruments?

L232 LADCP data were processes following thé inversion mehod of Visbeck (2002).

L236 The main instrument used was a SeaSoar ...

L238 a WET Labs WetStar chlorophyll-a fluorometer

L239 attached to ; profiled cable ?

L240 give scale in km

L242 range of sampling ; real-time

L244 please provide km too ; crossing numerous and various structures ... (see section 5), ... fine-scale patterns

L248 Avoid the use of underscore while refering to cruise name in the text and be consistent throughout the paper ; During the PROTEVS-MED 2017 cruise, a ...

L250 half a

L251 when the instrument is surfacing

L253 what is the conclusion of the comparison?

L258 allowed for real-time

L259 what is the horizontal resolution in km?

L263 what do you mean by "induct"? ; contributed

L271 on R/Vs Pourquoi pas ?, Atalante, Beautemps Beaupré ; are 150kHz and 38kHz Ocean Surveyor by RDI Teledyne

L291-292 ship CTD and SeaSoar ; SHOM? (please be consistent with capital letters) ; bath, whose temperature can ...

L298 with SBE 35's : do you mean by several SBE 35? In that case how many?

L304 tested against Autosal and Portasal salinometers.

L310 the provider ? You mean collaborators who provided the instruments ? An easy process would be to compare in the TS space the data from MVP and RapidScan with calibrated ship CTD casts...

L320 Level 0 (L0) consisting in ... Level 1 (L1) displaying ... standards units and corrected from eventual drift of sensors ... Level 2 (L2) proposed as ...

L334 and the corresponding author? Or provide an adresse/url to get the data. It would be nice to have a contact point without having to look for it.

L338 also 100m

L339 Lagrangian

L340 deployed with dedicated sampling rate. Please specify the range of temporal sampling, of the parking depth, and for how long.

L344 (see table 2)

L356 provided ... It was also possible to simultaneously observe.

L357 "showing the predominance ... (few kilomètres)." Without further demonstration, I would remove that strong and general assessment. (In general, please use past tense for data description and present for interpretation.)

L358 were patchier

L360 appeared ... described and are actually made of

L362 dual-core anticyclonic eddy was observed

L363 three-core eddy was also

L364 remains to be investigated

L365 pre-existing eddies

L366 reveal filaments and layered structures due to submesoscale (ageostrophic) dynamics

L367 symmetric instabilties -> what about stirring by the mesoscale eddy field, frontogenesis ...? There is no evidence here of one prevalent process... Please draw instead a list of relevant potential mechanisms.

L369 Northern ; a SCV of LIW was observed south of Toulon. Do you have evidence of swirling currents by VMADCP?

L371 was paid to the NC

L372 SCVs generated in the deep convection area.

L375 of the deep convection area

L376 showed small-scale structures likely formed by convection ; the north to the south

L378 convection chimney

L380 double-diffusive processes ; please cite existing references related to double-diffusion in the Western Mediterranean.

L382 Cap Creus

L384 cold water originating from the Gulf of Lion's shelf (please correct Gulf of Lion in future 1 too) ; The WIW was progressively entrained and mixed with the AW and LIW while flowing south ; Please add a word and a reference to dense shelf water cascading (e.g. Durrieu de Madron et al, 2013).

L397 the fine-scale dynamics

L401 It also complements the repeated glider lines maintained in the framework of the MOOSE observatory (Coppola et al, 2019) and is usefull to design future combined multi-plarform experiments.

L403 different instruments to obtain high-resolution

L404 transects have been

L405 has only been tested and used as ...

L409 perfect tool to identify mesoscale structures ...

L410 down to

L411 describe important surface and subsurface dynamical features.

L413 free-fall instrument, its setting is lighter

L415 equation and hydrography.

L417 of the sensors ... , this experiment of fast and ... revealed ... (Garreau et al, 2018)

L429-430 and the back and forth ... fruitfull. Not clear, please rephrase.

L421 knowledge of the mesoscale and smaller, filling some ...

L422 gains ; meters), sub-mesoscale ... starts to be resolved ... This dataset of observations at similar resolution ... simulated secondary circulations.

Figure 1 : you could consider putting names of main currents on the map for the reader who won't necessarily be familiar with the region.

Figure 2 : are there all the sensors used during the campaigns?

Table 2 : Argo floats ... deployed

Figure 3 : Dual-core observe in the NBF during XXX (name of cruise and year) ... Three-layer Eddy ... (Name of cruise instead of year) ... Front of XX -> observed off XX ... NC position and intensity (from which instruments and depth?) cross-front transect of the NBF ... double-diffusive staircases ... patch of cold ... Catalan Shelf ... in the data repository. Please also consider redrawing the figure, axis labels are lying ontop of each others.

Reference :

P Testor, JC Gascard : Large-scale spreading of deep waters in the Western Mediterranean Sea by submesoscale coherent eddies, Journal of physical oceanography 33 (1), 75-87, 2003

P Testor, JC Gascard, Post-convection spreading phase in the Northwestern Mediterranean Sea, Deep Sea Research Part I: Oceanographic Research Papers 53 (5), 869-893, 2006

Bosse, Circulation générale et couplage physique-biogéochimie à (sous-) mésoéchelle en Méditerranée Nord-Occidentale á partir de données in situ, PhD thesis, Sorbonne-Université, 2015

Giordani et al, A PV-approach for dense water formation along fronts: Application to the Northwestern Mediterranean, JGR, 2017

Medoc group, Observation of Formation of Deep Water in the Mediterranean Sea, 1969.ÂăNatureÂă227,Âă1037–1040 (1970) doi:10.1038/2271037a0

Schott et al, Observations of deep convection in the Gulf of Lions, northern Mediterranean, during the winter of 1991/92, JPO 1996

Coppola, L., P. Raimbault, L. Mortier, and P. Testor (2019), Monitoring the environment in the northwestern Mediterranean Sea,ÂăEos, 100,Âăhttps://doi.org/10.1029/2019EO125951

Durrieu de Madron et al, Interaction of dense shelf water cascading and open‐sea convection in the northwestern Mediterranean during winter 2012, GRL 2013

---

## Author Comment (AC1) · 21 Dec 2019

Thank you for your review and contribution. You will find hereafter in the Author Comments (ACs) our answers to the two reviewers comments and a tagged version of the revised paper.

Cordially Pierre Garreau

––––––––––––––––––––––––––––––––

---

## Author Comment (AC3) · 21 Dec 2019

Please find on the attached file our answers to the reviewers and a tagged change version of the revised manuscript.

Please also note the supplement to this comment: https://www.earth-syst-sci-data-discuss.net/essd-2019-173/essd-2019-173-AC3-supplement.pdf

---

## Author Response (AR1)

**Author's response to the comments on "PROTEVS-MED field experiments: Very High-Resolution Hydrographic Surveys in the Western Mediterranean Sea"**

**Pierre Garreau et al.**

**December 15, 2019**

**General response**

First we thank the two referees for their reviews and contributions. As these were minor revisions and as we are non native english speakers, we followed the recommendations of the referees in most cases. Hereafter, the detailed answers to the reviewer #1 and the reviewer #2.

The corrupted file (https://www.seanoe.org/data/00512/62352/data/66880.pdf) corresponding to the quicklooks of first leg of 2017 has been reprocessed, uploaded on the repository and is now downloadable. The full dataset has been also re-checked.

We were also contacted directly by the Seanoe administrator. He suggested to add the DOI of the GOSUD data repository in complement of the already cited paper; It has be added in the references.

**Reviewer #1 :**

**Line 24 "(about 10000 Km)" could be replaced by (total length about 10000 Km)**

Done.

**Line 28 TermoSalinoGraph (TSG) "CTD casts" could be replaced by Classical full depth CTD stations have been realized.... Please define CTD acronym. I think that the manuscript would benefit of a clear distinction between a CTD (instrument that is included in the towed system as well as in the free fall profiler) and a classical CTD cast performed when the ship stops in a sampling station. At the moment, in this manuscript the term CTD is used for all the classical oceanographic stations and this could generate some confusion.**

*Done. Thank you for this remark, we tracked CTD in the whole manuscript and removed the confusion between the instrument, the cast and the stations.*

**Line 29 "objects" may be replaced by "structures"**

done.

**Line 30 "the aim of the survey. . .." Please consider resentencing**

*As the main objectives of the surveys has been already defined few line above, this sentence has been removed. In there following statement 'nevertheless' has been also removed. We hope the text is now more fluent.*

**Line 32 "biological sensors. . ..have been carried out" Please consider resentencing**

*The concerned sentence has been rephrased as follow :*

*"When available, biological sensors (Chlorophyll a, Turbidity, Dissolved Oxygen etc.) have been carried out . They provided useful complementary observations about the circulation"*

**Line 44 "chlorophyll a " replace with chlorophyll a. Please be coherent throughout the paper**

*chlorophyll-a will be used for all the paper.*

**Line 47 "As the scales" might be replasec by "As all these scales" "to develop observations" please replace with"to develop an observation strategy"**

*done*

**Line 56-60 Please consider to move here the figure 1 also adding the geographical references mentioned in the text (up to section 3). I would also move to the very first lines the name of the study area.**

*The figure 1 has been split in two separate figures (figure 1 and figure 2) and a reference to the figure 1 has been added in this paragraph. Therefore the editor will probably put the figure 1 close to this paragraph. Crossed oceanographique features were added one the map*

 **Line 57 "depths under" maybe "below"**

*done*

**Paragraph 2.1 Please consider to add a figure showing the Mediterranean Sea and the oceanographic features described in this paragraph, as well as the location of the study area. Please add a description of the deep layer properties, or alternatively rename the paragraph to focus on surface and intermediate circulation and main water masses**

*See comment above, the figure 1 has been split in two part and repositioned in, the MS.*

*The paragraph has been changed in "Surface and intermediate circulation" as we focus on surface and sub-surface observations. If interested by the deeper water circulation, the reader can refer to the cited bibliography.*

**Line 101 "patchy ocean" maybe "patchy ocean areas"**

*Done. We agree, only the areas with strong scales interactions interactions exhibit possible patchy patterns.*

**Line 106 show**

*corrected*

**Line 112 because of**

*corrected*

**Line 121 high resolution in situ data by glider have also been compared to the new generation salinity products by SMOS satellite as in "Aulicino, G.; Cotroneo, Y.; Olmedo, E.;**

**Cesarano, C.; Fusco, G.; Budillon, G. In Situ and Satellite Sea Surface Salinity in the Algerian Basin Observed through ABACUS Glider Measurements and BEC SMOS Regional Products. Remote Sens. 2019, 11, 1361".**

*Thank you for this recent reference. It has been added in the text and in the references list.*

**Line 129 What you mean with "turning radius"? Is the ability to change direction?**

Yes. It is now precised in the sentence :

*"... this towed vehicle can handle a turning radius of 2 nautical miles (i.e. gyration speed of 10 degree/min), when the ship change direction."*

**Line 146 The parenthesis includes both oceanographic features that are described in the dataset and basins. Probably listing just one of the categories would be better.**

Done, only basins or geographical areas are now quoted.

**Line 148 "weddies" have not been defined before**

*This unique reference to weddies in our paper is replaced by the more explicit statement : "modal structures composed of WIW.*

**Line 155 "rapidcast" not mentioned or described before please add a description at lines 127-140 as for SeaSoar or MVP**

*The following sentence has been added in the abstract, close to the previous short description of the Seasoar and the MVP :*

*"In 2018, another free fall profiler (a RapidCast) has been tested."*

**Line 158 "CTD casts" maybe "classical CTD stations"**

*done*
**Line 160 "Shom" replace with "SHOM"**

done

**Line 165-166 Consider removing the inner parenthesis. i.e. "– VMADCP –"**

*done. Also corrected for (MVP) in the same way.*

**Line 192-195 NBF and NC have already been defined, please use the acronyms**

*done*

**Line 204 ". . .to a strong Mistral gust, part of the cruise. . .."**

*Done, coma is added.*

**Line 212 "lagrangian"**

*corrected*

**Line 217 "Finite Singular Lyapunov Exponents" consider adding a reference**

*Instead adding a theoretical paper, the reader can found in Nencioli 2018 for instance, I suggest the following example of the use of FSLE in Med Sea :*

d'Ovidio, F., V. Fernández, E. Hernández–García, and C. López (2004), Mixing structures in the Mediterranean Sea from finite–size Lyapunov exponents, Geophys. Res. Lett., 31, L17203, doi:10.1029/2004GL020328.

*This reference has been added in the text and in the reference list.*

**Line 244 "given" or "giving"?**

*Giving seems me better*

**Line 280 (latitude and longitude)**

*corrected*

**Line 292 "Shom" should be SHOM. Please check throughout the entire manuscript**

*Done and checked in the whole ms.*

**Line 294 in the case. . ..which is the more common"**

*corrected :*

*in the case. . ..which is the most common*

**Line 298 "SBE 35's" maybe "SBE 3's"?**

*Unchanged:  SBE 35  is a temperature sensor ideal for use in calibration labs.*

**Line 319 "available for data" maybe "available for each dataset"**

*yes, thank you, we agree. And the sentence has been modified.*

**Line 320 "consists in" maybe "consists in the" "sensor" should be sensors"**

*done*

**Line 325 Are the gridded profiles averaged along depth only? The term gridded may confuse the reader**

*profiles are only vertically resampled and positioned as vertical profile at the mean geographical position of the considered record.The comment is now rewritten as :*

*"The third level (Level 2, L2) is proposed as gridded, controlled and resampled data in netcdf files (.nc). Gridded dataset for salinity and temperature have been vertically resampled every meter removing spike, spurious values, density inversion when they persist after the first process supplied by the sensor manufacturer. They are then positioned as vertical profiles at the mean geographical position of the considered up or down record"*

**Line 340 What you mean with "higher temporal resolution"? Please clarify. If the float is enveloped in a structure it would provide longer observations in time.**

*We hope this new text will be more clear.*

*"Few Argo floats were also dropped and experienced first Protevsmed dedicated mission with high temporal resolution (daily cycle) and a parking depth adjusted to the observations to maintain the drifters as long of possible in the targeted structures (typically 100 m deep).  When the drifter left*

*the structure, it used the usual Argo standard procedure in the Mediterranean (i.e. a 5-day cycle and a parking depth of 350m)."*

**5. Overeiw of the observations As stated in the introduction, here a limited number of sample analysis of the collected data are offered to the reader. Please consider adding a sentence at the beginning of the chapter that clarify this.**

*In the introduction we replaced some "quicklooks" by "an overview".*

*The chapter 5 has been changed :*

*"5 Overview of selected observations"*

*Please note that in the text and in the figure caption the availability of extensive quicklooks on the repository is quoted.*

**Line 356 "When deployed"**

*Corrected*

**Line 376 "frequently show"**

*Corrected*

**Line 379 "CTD casts" should be "classical CTD stations"**

*Done.*

**Line 379-381 Please add the position of these casts on figure**

*The exact position of the casts are now given in the figure caption.*

**Figure 3 Please consider splitting this figure in order to obtain an higher definition for each plot**

*The aim of this figure and to provide an overview of the results at a glance. We have chosen to keep it. Nevertheless, a more higher resolution version will be proposed for the final version of the paper as the figure have to be uploaded separately*

**Line 398-401 Please consider resentencing Line 404 "Some transects have been"**

*corrected*

**Reviewer #2 :**

**L25 missing ,**

*Corrected*

**L29 "peculiar objects" prefer particular processes ; a MVP**

*Corrected following the first reviewer suggestion:*

*"peculiar structures"*

**L30-32 ... to access even higher resolution the ocean physics (temperature, salinity, currents). Biological sensors were opportunistically used to provide ...**

The concerned sentence has been rephrased as follow:

*"When available, biological sensors (Chlorophyll a, Turbidity, Dissolved Oxygen etc.) have been carried out . They provided useful complementary observations about the circulation"*

**Throughout the paper : opportunely -> opportunistically**

*Ok ,all occurrence of "opportunely" were checked and as opportunistically seems to be pejorative, I prefer to remove this adverb in the paper.*

**L33 the fine-scale processes**

*corrected*

**L35 properly résolve**

*corrected*

**L43 connecting ... to ; energy cascades to small scales and reversely**

*corrected*

**L44 For instance in the northwestern Mediterranean ; chlorrophyll-a**

corrected

**L46 vortices**

*done*

**L50-55 partly fill these gaps between the large and finer scale dynamics. Since many years, remotely ... observe a large ... soon by the future Surface Water and Ocean ... mission (SWOT), expecting to provide ... small-scale processes ...**

*thank you for rephrasing*

**L59 in a context of ocean physics I would use km instead of nautical miles.**

*done*

**L63 looks at ; are described**

*following a request of the first reviewer, "quick looks" has been replaced by "overview".*

**L66 Oceanic context of the**

*corrected*

**L68 is sometimes referred as a "lab-ocean"**

*unchanged : "pocket ocean" is also used.*

**L70 "nearby" did you mean close to Land? Accessible?**

*Accessible seems the better word.*

**2.1 General circulation : you aren't describing the thermohaline circulation very much...**

*Yes we focus only on the surface and intermediate circulation. The title has been changed as suggested by the first reviewer.*

**L74 sub-basin circulation ; what to you mean about dominated? Please rephrase.**

*The new sentence is :*

*"The basin or sub-basin dynamics is largely driven by the thermohaline circulation."*

**L75 basin or sea, not both ; Light (fresh) Atlantic Water**

*corrected*

**L76 generally circulates along the continental slope ... western and eastern basins**

*corrected as suggested*

**L77 The slope ... Algerian coast**

*done*

**L79 and throughout the paper, no caps for western basin etc**

*ok, we checked and corrected all occurrences*

**L77-79 These lines fit better to 2.2**

*Yes, but we can't evoke the Algerian current without is instabilities.*

**L79 northern part**

*done*

**L80 Ligurian Sea ; Northern Current is more often found (please modify consistently throughout the paper).**

*ok, we checked and corrected all occurrence*

**L83 (sounds more logic in this order) The generally accepted concept of the Northern Gyre flowing cyclonically around the doming of isopycnals of the deep convection area of the northwestern Mediterranean (please cite a ref about the general circulation). The existence, position and strenght of the return flow of this gyre ...**

*ok we adopt your formulation :*

*"The generally accepted concept of the Northern Gyre flowing cyclonically around the doming of isopycnals of the deep convection area of the northwestern Mediterranean Sea. The existence, position and strength of the return flow of this gyre is still under debate."*

*The well-established publications on circulation in the Western Mediterranean have already been widely cited in this paragraph and are not useful after all the statements.*

**L84 (LIW), a mode water ...basin entering the western ... , follows pattern (again please provide ref)**

*corrected and (Millot and Taupier Letage, 2005) ref is added.*

**L87 This important water mass is marked by a relative ...**

*modified as suggested*

**L93 indicator of ; because the scale of surface-intensified eddies in geostrophic bal- ance ranges a few ...**

*modified as suggested*

**L95-96 topography ; whose size is close to the local ; observed in the Western Mediterranean (also cite Testor and Gascard 2003, 2006; Bosse et al 2015, 2016)**

*citation added and text corrected.*

**L97 contrasted in terms of what? What's the point of reference? Subtropical regions might not be comparable, while polar regions would tend to exhibit similar characteristics with density compensated contrasts.**

*Ok, this part of the sentence was not useful* and *lead to confusion, and has been removed*

**L99 Submesoscale (tends to be found in one word in the litterature, please correct throughout the paper)**

*ok, submesocale has been adopted for the whole paper (except in cited references)*

**L101 marked by frequent strong wind évents ... northwestern basin ... with the NC and**

**mesoscale structures and generate ... (No need for PV acronym if not used afterward) ... please the relevant litterature : Bosse 2015, Estournel et al 2016, Giordani et al 2017, Testor et al 2018**

*references added*

**L104 a place ; for a long time (cite for instance MEDOC group et al 1979, Schott et al 1996, Houpert et al 2016, Testor 2018)**

*citation added*

**L106 both models ... and observations ... show**

*corrected*

**L108 due to the deepening of the mixed layer**

*ok, "mixed layer" is more appropriate*

**L111-115 ... are only partially resolved by usual ... ; repeated glider lines (instead of glider fleet) ... in the Western Mediterranean, in particular as part of the multi-platform ... MOOSE (rather cite Coppola et al 2019). Intensive targeted ... the dynamics of the deep convection area in the northwestern Mediterranean Sea (Estournel et al 2016).**

*Thank you for this very recent synthetic publication about MOOSE.*

*Repeated glider lines are discussed below.*

**L116 glider lines**

*done*

**L117 opportunely**

*removed ; see remark above ( L30-32)*

**L121-122 please sort references in chronological order**

*Done*

**L126 prefer 15-30 km/day than nautical miles**

*Done*

**L130 sampling of**

*Done*

**Proposition of section/subsection titles: 3 The PROTEVS-MED field experiments 3.1 Objectives 3.2 Cruises**

*Unchanged; We prefer our organisation.*

**L146 key regions of the basin ; North Balearic Front**

*Done. We took also into account the remarks of the first reviewer.*

**L147 "assessments of numerical simulation" is not very clear... What kind of simulation? How?**

*The text has been re-sentenced as follow :*

*"The goal was the assessments of operational numerical simulation of the circulation performed for the Navy."*

**L148 please avoid modal weddies which is not defined earlier ; surface and subsurface mode water eddies (including SCV)**

*Following also the comment of reviewer #1 the text has rephrased as follow :*

*"- to identify and follow peculiar mesoscale structures such as surface eddies, modal structures composed of WIW, submesoscale coherent vortices (SCV) meanders and filaments and explore their signatures on the sea surface height (altimetry) and their acoustic impact (i.e. through their modulation of the sound propagation speed)."*

**L149 meanders and filaments**

*Done, see above.*

**L151 observe and characterize**

*Done*

**L155 prefer platform to vectors... Also in other occurrences in the paper L158 across strong thermocline ; shipborne CTD casts.**

*Done and checked for the whole manuscript.*

**L161 SHOM?**

*Caps are now used for SHOM in the whole manuscript (following also the comment of reviewer #1)*

**L164 ship CTD**

*TSG is an appropriate term for ship CTD (see for example Gaillard et al., 2015). Nevertheless for this occurrence of the acronym in the MS, the sentence has been modified as follow :*

*"Ship board routinely acquired data (Vessel Mounted Acoustic Doppler Current Profiler - VMADCP- and ThermoSalinoGraph -TSG-) were also included in this database"*

**L166 ))**

*corrected*

**L168 dataset of all**

*corrected*

**L171 to detect and track mesoscale structures during the cruises.**

*We agree, suggested precisions were added*

**L178 paid to cross-slope transects**

*Corrected*

**L180 please refer to table 2. Apparently surface drifters were deployed during every cruise. Since it is not only the case for the first one, it feels odd to mention it here and not for the others.**

*The use of surface and argo drifters is now quoted at the end of the previous paragraph. Therefore the mention is here removed.*

**L183 Balearic Sea (please check capital letters for the rest for the rest of the paper).**

*Done*

**L185 therefore led ; mostly carry out ship CTD casts**

*Corrected ; CTD casts has been replaced by CTD stations following the comment of reviewer #1*

**L188 with caution, as they caused an excessive ...**

*corrected*

**L193 origin of the NC where the flow through the Corsica Channel and the WCC join.**

*Modified as suggested*

**L194-195 This latter ... current. I am not sure about what the authors are trying to say here.**

*We hope the following sentence is more clear:*

*The behaviour and the origin of the WCC was also explored along the western coast of Corsica.*

**L195 in early spring**

*corrected*

**L196 to capture an Algerian Eddy**

*corrected*

**L198 also provides insights about ; Northern Current**

*corrected*

**L203 eddy tracking tool**

*corrected*

**L204 consecutive to a strong Mistral gust, …**

*corrected*

**L206 (WIW) formation were ... ; A SCV was**

*corrected, WIW acronym is now defined above and the repetition of "water" also is removed.*

**L212 release**

*corrected*

**L213 what is the acronym of SPASSO?**

*SPASSO acronym is quickly available on the associated web site. Moreover the SPASSO functions are described in the sentence.  Therefore we remove SPASSO from the text.*

**L215 mushrooms-like structures -> do you mean dipolar structures ? ; fronts ; focus was given to the area south of Mallorca where**

*done (3 corrections)*

**L216 a front was detected ; Lagrangian diagnosis**

*corrected*

**L218 "A Lagrangian round-trip strategy" I don't know what this is...**

*Surface drifters were dropped at biological key points in the area and revisited few time a day by the ship in order to follow the phyto- and zooplankton during a diurnal cycle following the so marked water mass. As this strategy concern the biological part of the cruise, not included in this data paper we chose to simplify the sentence as follow :*

*A Lagrangian strategy was specifically set up in order to study the structure and growth rate (at 24 hours time scale) of the various phytoplankton groups as defined by flow cytometry measurements as in Marrec et al. (2018).*

*A full description of the strategy is expected in another paper devoted to the biological results.*

**L230 a RDI 150khz ... Was the LADCP profiles acquired by a pair of those instruments?**

No,  the rosette was equipped only with one ADCP. *The text ins therefore unchanged*

**L232 LADCP data were processes following thé inversion mehod of Visbeck (2002).**

*Modified , thank you for the precision.*

**L236 The main instrument used was a SeaSoar ...**

*Already indicated in the §3. We prefer our redaction.*

**L238 a WET Labs WetStar chlorophyll-a fluorometer**

*corrected*

**L239 attached to ; profiled cable ?**

*The concerned sentence has been simplified as follow ;*

*"The SeaSoar was trawled at 9 knots by a profiled cable""*

**L240 give scale in km**

*done 1 nautical mile ~2 km*

**L242 range of sampling ; real-time**

*corrected*

**L244 please provide km too ; crossing numerous and various structures ... (see section 5), ... fine-scale patterns**

*done 5500 nautical miles ~ 10.000 kilometres*

*modified following suggestions.*

**L248 Avoid the use of underscore while refering to cruise name in the text and be consistent throughout the paper ; During the PROTEVS-MED 2017 cruise, a ...**

***Ok, checked for all the MS***

**L250 half a**

*corrected*

**L251 when the instrument is surfacing**

*corrected*

**L253 what is the conclusion of the comparison?**

**L258 allowed for real-time**

*corrected*

**L259 what is the horizontal resolution in km?**

*With a resolution similar to the seasoar (about 2 km)*

**L263 what do you mean by "induct"? ; contributed**

*"induct" is not the correct word. "Inlet" is now used. See Gaillard et al (2015) for complement about the protocol.*

*Contributed* **:** *corrected*

**L271 on R/Vs Pourquoi pas ?, Atalante, Beautemps Beaupré ; are 150kHz and 38kHz Ocean Surveyor by RDI Teledyne**

*corrected*

**L291-292 ship CTD and SeaSoar ; SHOM? (please be consistent with capital letters) ; bath, whose temperature can ...**

*no, it's mean all CTD used during the cruises, the "all" is added in the sentence.*

*Shom : corrected*

*bath, whose : corrected*

**L298 with SBE 35's : do you mean by several SBE 35? In that case how many?**

*The "s" is removed. The SHOM labs is equipped with 2 baths and 3 SBE 35's but the common protocol request only one SBE 35.*

**L304 tested against Autosal and Portasal salinometers.**

*Corrected as suggested*

**L310 the provider ? You mean collaborators who provided the instruments ? An easy process would be to compare in the TS space the data from MVP and RapidScan with calibrated ship CTD casts...**

*Unfortunately, we don't have an access to any pre- or post-calibration for the MVP or the RapidCast sensors. The only one calibration we had was the one provided by the constructor. Of course a comparison with other measurement in the vicinity remain possible. Note, that as mentioned in the MS, we focus more on the structures than on absolute values.*

**The following sentence is now added :**

*but a comparison with the results of calibrated SBE sensors can be carried out..*

**L320 Level 0 (L0) consisting in ... Level 1 (L1) displaying ... standards units and corrected from eventual drift of sensors ... Level 2 (L2) proposed as ...**

*done*

**L334 and the corresponding author? Or provide an adresse/url to get the data. It would be nice to have a contact point without having to look for it.**

*An email address is provided : data-support@shom.fr*

**L338 also 100m**

*Right, 2 drifters with holey socks positioned at 100 m were dropped in 2018. modified in the MS*

**L339 Lagrangian**

*corrected*

**L340 deployed with dedicated sampling rate. Please specify the range of temporal sampling, of the parking depth, and for how long.**

*Done, information also requested by the reviewer #1; the new redaction is :*

*"Few Argo floats were also dropped and experienced first PROTEVS-MED dedicated mission with high temporal resolution (daily cycle) and a parking depth adjusted to the observations to maintain the drifters as long of possible in the targeted structures (typically 100 m deep). When the drifter left the structure, it used the usual Argo standard procedure in the Mediterranean (i.e. a 5-day cycle and a parking depth of 350m)."*

**L344 (see table 2)**

*modified as suggested*

**L356 provided ... It was also possible to simultaneously observe.**

*Corrected, thank you for the tense correction.*

**L357 "showing the predominance ... (few kilomètres)." Without further demonstration, I would remove that strong and general assessment. (In general, please use past tense for data description and present for interpretation.)**

*OK, the sentence is rephrased in a less strong and more general way. Nevertheless this observation is important in the framework of the future SWOT program. Even at relatively small scale (in the range of 10 km) the dynamics follow mainly the geostrophic (or cyclogeostrophic) balance. The new redaction is now:*

*... "showing the importance of the geostrophic balance even at small scale (in the range of 10 km)"*

**L358 were patchier**

*corrected*

**L360 appeared ... described and are actually made of**

*tense is now corrected. Commonly is preferred to actually in the new redaction.*

**L362 dual-core anticyclonic eddy was observed**

*corrected*

**L363 three-core eddy was also**

*modified as suggested*

**L364 remains to be investigated**

*modified as suggested*

**L365 pre-existing eddies**

*modified as suggested*

**L366 reveal filaments and layered structures due to submesoscale (ageostrophic) dy- namics**

*We prefere our sentence; not modified.*

**L367 symmetric instabilties -> what about stirring by the mesoscale eddy field, fronto- genesis ...? There is no evidence here of one prevalent process... Please draw instead a list of relevant potential mechanisms.**

*Frontogenesis is added. Stirring is evoked in the next sentience.*

**L369 Northern ; a SCV of LIW was observed south of Toulon. Do you have evidence of swirling currents by VMADCP?**

Yes*. **The sentence are now :***

"In the North Current, stirring appeared in both tracers and velocity fields and an SCV formed by LIW detached in front of Toulon (figure 3c) was observed as confirmed by observed swirling velocities on VMDCP records. It was also topped by a surface cyclonic gyre."

**L371 was paid to the NC**

*tense corrected*

**L372 SCVs generated in the deep convection area.**

*Ok we agree*

**L375 of the deep convection area**

*ok geographical position is not useful. removed*

**L376 showed small-scale structures likely formed by convection ; the north to the south**

*corrected as suggested*

**L378 convection chimney**

*corrected as suggested*

**L380 double-diffusive processes ; please cite existing references related to double- diffusion in the Western Mediterranean.**

*(Onken, R., Brambilla, 2003) is added*

**L382 Cap Creus**

*corrected as requested*

**L384 cold water originating from the Gulf of Lion's shelf (please correct Gulf of Lion in future 1 too) ; The WIW was progressively entrained and mixed with the AW and LIW while flowing south ; Please add a word and a reference to dense shelf water cascading (e.g. Durrieu de Madron et al, 2013).**

*modified as suggested*

*Clearly it is not shelf water cascading as described by Durrieu de Madron, but a density adjustment of a water masses that will probably form WIW. Nevertheless the suggested reference is added to highlight the differences in  the processes.*

**L397 the fine-scale dynamics**

*modified as suggested, very is removed*

**L401 It also complements the repeated glider lines maintained in the framework of the MOOSE observatory (Coppola et al, 2019) and is usefull to design future combined multi-plarform experiments.**

*modified as suggested after typing errors corrections.*

**L403 different instruments to obtain high-resolution**

*modified as suggested*

**L404 transects have been**

*corrected*

**L405 has only been tested and used as …**

*modified as suggested*

**L409 perfect tool to identify mesoscale structures ...**

*ok, drive out is replaced by identify*

**L410 down to**

*modified as suggested*

**L411 describe important surface and subsurface dynamical features.**

*modified as suggested*

**L413 free-fall instrument, its setting is lighter**

*modified as suggested*

**L415 equation and hydrography.**

*modified as suggested,*

**L417 of the sensors ... , this experiment of fast and ... revealed ... (Garreau et al, 2018)**

*modified as suggested,*

**L429-430 and the back and forth ... fruitfull. Not clear, please rephrase.**

**May be synergy is more a more conceptual formulation. The new versionof the sentence is now :**

In situ observations of ageostrophic dynamics remain rare and the synergy between these observations and theory, and then between these observations and modelling, should be very fruitful.

[revised manuscript text omitted]

---

## Author Response (AR2)

Dear Editor,

Thank you for your suggestions and precisions.

**Para2 line 76. The Mediterranean Sea is ... a 'pocket ocean'. It is written 'often referred to', but in the added reference (Robinson et al 2001) this saying is not found. On the contrary, the Mediterranean is defined as a 'laboratory basin' in the book by Malanotte Rizzoli and Robinson (editors) Ocean Processes in Climate Dynamics: Global and Mediterranean Examples. Authors should adopt this terminology**

We checked the reference and we use now the exact wording of Robinson: "laboratory basin" and have also changed the reference: (Robinson and Glonaraghi, 1994);

Robinson, A.R., Golnaraghi, M., 1994. The Physical and Dynamical Oceanography of the Mediterranean Sea, in: Malanotte-Rizzoli, P., Robinson, A.R. (Eds.), Ocean Processes in Climate Dynamics: Global and Mediterranean Examples. Springer Netherlands, Dordrecht, pp. 255–306. https://doi.org/10.1007/978-94-011-0870-6\_12

**Para 2.1 line 95 - 'Levantine Intermediate Water (LIW) which is a modal water'. LIW is a 'water type' (historical reference of water type is the well-known book The Oceans: Their Physics, Chemistry, and General Biology Sverdrup, H.U., Johnson, M.W. & Fleming. This nomenclature is in accordance with the sentence (line 99) 'This important water mass is marked by a relative subsurface maximum of temperature and salinity'**

The reference to Sverdrup et al. 1942 has been also added when the characteristics of this water masse in the western basin is discussed. To remove any ambiguous formulation, any referring to modal water is removed. The new redaction is:

"The Levantine Intermediate Water (LIW) formed in winter in the eastern basin, entering into the western basin through the Sicilian Strait, follows more or less the same cyclonic circulation pattern (Millot and Taupier Letage, 2005). It spreads out into the northern part of the western basin between 400 and 800 m depths and is found sporadically within the Algerian Basin. In the western basin, this important water mass is marked by a relative subsurface maximum of temperature and salinity was already quoted as water type in Sverdrup et al. (1942)."

Note that in Sverdrup (1942) this water mass has been described only as "Intermediate Water".

**Please find hereafter the tracked change version of the manuscript.**

Cordially

Pierre Garreau

**PROTEVS-MED field experiments: Very High-Resolution Hydrographic Surveys in the Western Mediterranean Sea**

**5**

Pierre Garreau1, Franck Dumas2, Stéphanie Louazel2, Stéphanie Correard2, Solenn Fercoq2, Marc Le Menn2, Alain Serpette2, Valérie Garnier1, Alexandre Stegner3, Briac Le Vu3, Andrea Doglioli4, Gerald Gregori4

**10**

[revised manuscript text omitted]